# Enhancing Embedded Object Tracking: A Hardware Acceleration Approach for Real-Time Predictability

**DOI:** 10.3390/jimaging10030070

**Published:** 2024-03-13

**Authors:** Mingyang Zhang, Kristof Van Beeck, Toon Goedemé

**Affiliations:** PSI-EAVISE Research Group, Department of Electrical Engineering, KU Leuven, 2860 Sint-Katelijne-Waver, Belgium; kristof.vanbeeck@kuleuven.be (K.V.B.); toon.goedeme@kuleuven.be (T.G.)

**Keywords:** deep learning, object tracking, siamese network, FPGA, real-time system predictability, hardware acceleration, high-level synthesis, embedded system

## Abstract

While Siamese object tracking has witnessed significant advancements, its hard real-time behaviour on embedded devices remains inadequately addressed. In many application cases, an embedded implementation should not only have a minimal execution latency, but this latency should ideally also have zero variance, i.e., be predictable. This study aims to address this issue by meticulously analysing real-time predictability across different components of a deep-learning-based video object tracking system. Our detailed experiments not only indicate the superiority of Field-Programmable Gate Array (FPGA) implementations in terms of hard real-time behaviour but also unveil important time predictability bottlenecks. We introduce dedicated hardware accelerators for key processes, focusing on depth-wise cross-correlation and padding operations, utilizing high-level synthesis (HLS). Implemented on a KV260 board, our enhanced tracker exhibits not only a speed up, with a factor of 6.6, in mean execution time but also significant improvements in hard real-time predictability by yielding 11 times less latency variation as compared to our baseline. A subsequent analysis of power consumption reveals our approach’s contribution to enhanced power efficiency. These advancements underscore the crucial role of hardware acceleration in realizing time-predictable object tracking on embedded systems, setting new standards for future hardware–software co-design endeavours in this domain.

## 1. Introduction

The realm of object tracking has undergone a significant transformation in recent years, driven predominantly by the advent of deep learning techniques. Among the various approaches, Siamese networks have emerged as a leading solution, proficiently addressing the complexities of object appearance, scale variations, and occlusions [1]. Originating in the domain of image recognition and classification, these networks have been adeptly adapted to the dynamic requirements of object tracking. However, their deployment in real-world applications, particularly on embedded devices, poses substantial challenges. Many application cases require a hard real-time system, that is designed to meet strict time constraints, with a guaranteed response time for its critical tasks. A hard real-time tracker should thus have a small execution latency for each video frame, which moreover ideally is a constant, non-varying number. The real-time predictability of deep learning-based trackers, especially those initially designed for desktop GPU environments, remains a critical concern. Field-Programmable Gate Arrays (FPGAs) offer a promising solution, leveraging their parallel processing capabilities, efficient resource utilization, and reconfigurability to boost performance and minimize the power consumption of neural networks compared to conventional CPUs and GPUs [2]. While existing studies have delved into the FPGA acceleration of neural networks, the specific analysis of real-time predictability in such implementations has often been neglected. This gap underscores the urgent need for comprehensive research into FPGA-accelerated Siamese trackers, especially focusing on integrating complex features, as demonstrated by advanced models like SiamRPN++ [3].

Addressing this gap, our paper presents a dual-faceted approach. Firstly, we present a novel, detailed analysis of the real-time predictability of components within a deep learning-based object tracking system. This analysis aims to identify and address the bottlenecks that impede real-time performance. Based on our findings, we subsequently propose the design of two dedicated hardware accelerators, specifically tailored to alleviate the identified bottlenecks. This approach represents a significant shift from conventional methodologies, which often prioritize accuracy and tracking performance without due consideration for real-time operational constraints.

Our solution leverages the synergy of a heterogeneous system, combining the versatility of CPU processing with the parallel processing prowess of FPGA technology. This combination is particularly effective in enhancing the real-time predictability of the tracking system, which is a crucial requirement for many practical applications. The experimental results presented in this paper not only demonstrate improvements in runtime efficiency but also, and perhaps more importantly, mark significant strides in achieving real-time predictability. These improvements are especially pronounced with the integration of the dedicated hardware accelerators.

Moreover, an integral aspect of our research delves into the power consumption of these embedded systems. By measuring and analysing the power efficiency under various hardware configurations, our study brings to light the energy aspects of hardware acceleration in object tracking. This dimension of power efficiency is critical in embedded systems. Our findings in this regard offer valuable insights into power usage optimization, thus contributing to the development of more sustainable and energy-efficient tracking systems.

In summary, this work shines a light on the attainable synergy between hardware acceleration and real-time object tracking by addressing the often-overlooked aspects of predictability in related works. By demonstrating substantial improvements in mean speed, speed variance, and worst-case speed by factors of 6.6, 11.5, and 6.9, respectively, our methodology showcases a promising blueprint for refining real-time performance in the realm of object tracking. It sets new standards in the efficient and predictive implementation of Siamese networks in embedded systems, potentially paving the way for future research focused on balancing performance, predictability, and power efficiency in real-time applications.

## 2. Related Work

Visual tracking, a crucial subfield of computer vision, has experienced substantial advancements over the years. This section revisits the key milestones in object tracking, highlighting the evolution from traditional methods to deep learning-based approaches, and focuses on the role of hardware acceleration in embedded systems.

### 2.1. Object Tracking

Visual tracking began with classical methods, such as the Mean Shift [4,5,6] and Kalman Filter [7,8,9] methods, relying on colour histograms and motion prediction. Despite their foundational impact, these techniques often struggled in complex scenarios, such as those involving occlusions and rapid motion changes.

The integration of machine learning brought forth methods like TLD [10] and Struck [11], which combined tracking and detection mechanisms, improving adaptability to changes in object appearance. The major transformation, however, occurred with the advent of deep learning in tracking. CNNs, as seen in MDNet [12] and GOTURN [13], significantly advanced feature extraction and object localization in tracking. MDNet’s multi-domain learning and GOTURN’s regression-based approach marked critical improvements. The use of RNNs, particularly LSTM units in ROLO [14] and GANs in SINT++ [15], further diversified deep learning applications in tracking, enhancing their capability in complex visual scenarios.

Siamese networks, introduced in SiamFC [16], have been particularly effective in object tracking. Employing a novel cross-correlation layer, this architecture balances accuracy and speed. The SA-Siam [17] tracker learns semantic and appearance features in two dedicated branches to keep the heterogeneity of the two types of features. The SiamRPN tracker, introduced in [18], incorporates a region proposal network (RPN), as proposed in [19], into the Siamese architecture, thereby improving the accuracy of bounding box prediction. The SiamRPN++ tracker [3] introduces several complex features to extend the capabilities of the SiamRPN tracker. Notably, it employs ResNet50 [20] as the backbone network, which is a departure from the conventionally used AlexNet [21] in previous works. This deeper and more complex network enhances the extraction of richer and more discriminative features, thereby boosting the tracker’s adaptability and accuracy across diverse tracking scenarios. The SiamRPN++ tracker also incorporates feature aggregation from various backbone levels. This multi-level feature aggregation allows the tracker to collect a broader spectrum of object information, including high-level semantic features and low-level texture details. To maximize the utility of the features extracted from different layers of ResNet50, the authors introduced depth-wise cross-correlation, which generates a fine-grained similarity map for more precise tracking.

Recent progress in object tracking has witnessed a notable shift towards transformer-based methods, as these models excel in capturing complex spatial and temporal dynamics. Trackers like STARK [22], TrDiMP/TrSiam [23], TransT [24], and others, following the leads of ViT [25], DETR [26], and Swin Transformer [27], have adopted transformers to enhance tracking capabilities. These trackers utilize the transformer’s powerful feature extraction and context comprehension capabilities. However, the practical deployment of these sophisticated models on embedded systems faces significant challenges. Their high computational requirements and the nascent stage of transformer model quantization research make them less suitable for low-power, embedded applications where real-time processing and energy efficiency are paramount.

### 2.2. Embedded System Acceleration

While Siamese trackers have shown significant advancements in object tracking accuracy, their deployment on embedded devices, and particularly in real-time systems, is an aspect that still warrants exploration. Real-time systems are characterized by their need for both logical correctness and timely response. In these systems, predictability, which ensures consistent performance, is critical.

The existing literature includes several works that have taken strides in this area. For example, a notable work is [28], where the authors introduced a refined Siamese tracker based on SiamFC, termed MiniTracker. This tracker with post-training network quantization and pruning was deployed on a ZedBoard with a ZCU102 core, with FPGA resources manually allocated to different parts of the MiniTracker accelerator. However, the discussion surrounding the preprocessing and postprocessing stages, which typically run on the CPU, was notably absent in this work. This omission is significant as these stages could potentially impact the tracker’s real-time predictability.

Similarly, in [29], the authors optimized a SiamFC-like Siamese tracker for FPGA implementation. They evaluated the impact of quantization settings and backbone architecture on tracking accuracy, thus providing insights into the effect of quantization on the performance of Siamese Neural Networks. However, the precise architecture of the tracker was not clearly articulated, and the quantized tracker was not implemented on an actual FPGA, thereby complicating the reproduction and further development of other Siamese trackers.

The authors of [30] developed a tiny network with low computation requirements to enhance real-time performance with an acceptable accuracy drop. However, their tracker was not implemented on embedded devices, but instead on a desktop GPU.

In [31], a Siamese tracker based on the PYNQ framework was implemented on a ZCU104 board. Two IP cores were designed in this heterogeneous system to accelerate the Siamese network and the RPN in programmable logic (PL). The paper, however, did not clearly describe the Siamese tracker’s exact structure or the quantization setting used, making reproduction or improvement of the results challenging.

The authors of [32] introduced a hardware–software implementation of SiamFC on the ZCU104 platform. Their work examined the impact of quantization and parallelization settings on the accuracy and speed of the SiamFC implementation on FPGA. They provided a timing analysis of the tracker’s different components, a detail absent in previous works. However, this analysis did not include the maximum value or the standard deviation of execution time, which are critical metrics indicating the predictability of a real-time system.

Several gaps are evident in the existing literature on Siamese trackers’ hardware acceleration. Previous works focused on implementing SiamFC or similar trackers, neglecting recent features that enhance tracker performance, such as bounding box regression or feature aggregation from different levels. While these works claimed to improve real-time performance, they did not provide evidence of improved predictability in the real-time tracking system.

## 3. Real-Time Performance Analysis

In this section, we analyse the real-time predictability of the quantized SiamRPN++ tracker implemented on an embedded system, focusing on identifying potential bottlenecks. The results of our experiments serve as a baseline for further enhancements.

### 3.1. SiamRPN++ Tracker

As noted in Section 2, previous hardware implementations of Siamese trackers often lack complex components such as deep backbones, multi-layer features, and bounding box regression. Recognizing this gap, we chose the SiamRPN++ [3] tracker for its inclusion of these features, which not only enhances performance but also adds complexity to the hardware implementation.

The SiamRPN++ architecture employs a ResNet50 [20] backbone, as shown in Figure 1, for feature extraction from template and search frames. The ‘neck’ part processes these features, ensuring dimension uniformity, while the ‘head’ conducts depth-wise cross-correlation before producing classification and bounding box regression maps.

Depth-wise cross-correlation, a vital component in object tracking, reduces computational demands by operating independently on each channel. This efficiency is essential for real-time tasks and simultaneously improves tracking accuracy by capturing semantic correlations between features. A comprehensive mathematical analysis of this process will be detailed later.

Utilizing the generated similarity map, SiamRPN++ creates a classification score map and a bounding box regression map. During the prediction phase, these maps assist in identifying the target object’s location in the search image, combining the highest scoring area from the classification map with the corresponding adjustments from the regression map.

### 3.2. Quantization of the Tracker

The first step in implementing the SiamRPN++ tracker on embedded hardware is its quantization. FINN [33] has been the go-to FPGA acceleration automation framework for deploying Siamese trackers on FPGA. In FINN’s approach, each layer of the network intended for deployment is transformed into a separate component. The final network is then an IP composed of these individual components. Vitis-AI [34], also from Xilinx, offers an alternative approach, leveraging accelerators with an overlay-style architecture as opposed to the dedicated circuit architecture found in FINN. The process of implementing a PyTorch model in FINN becomes complex, requiring the model to undergo a Quantization-Aware Training (QAT) process with Brevitas [35] (a PyTorch library for neural network quantization). In contrast, Vitis-AI, with its use of Deep Learning Processor Unit (DPU) [36] IP (a programmable engine dedicated for convolutional neural network), does not require such a complicated process. Thus, we opted for Vitis-AI, given our primary aim to enhance real-time predictability rather than accuracy or latency performance on FPGA.

We employed Post-Training Quantization (PTQ) using Vitis-AI for the trainable parts of SiamRPN++ (i.e., backbone, neck, and head). This process is intended to efficiently balance computational resources and quantization accuracy. The main configurations of the quantization are detailed in Table 1. All data types in the model, including input, weights, bias, and activation, were quantized. All data types use the symmetric mode, which ensures that the distribution of quantized values is symmetric around 0. This feature is critical for reducing the quantization error and thereby improving the model’s accuracy. The ‘diffs’ method is used for all data types. This refers to the quantization algorithm used by Vitis-AI, which helps to reduce quantization loss and improve the performance of the quantized models. The round method is ‘std_round’ for input, weights, and bias and ‘half_up’ for activation. This method determines how rounding is performed during quantization. All data types are quantized at ‘per_tensor’ granularity, meaning that a single scale and zero point are used for all values within each tensor. The ‘poweroftwo’ scale type is used for all data types. This implies that the scale used for quantization is a power of two, which is more efficient for DPUs.

After quantization using Vitis-AI, we observed a notable efficiency in the process. Since the high tracking accuracy was not the main focus, we avoided the more complex QAT and opted for 8-bit PTQ. The decision to use the VOT2018 [37] dataset for our final evaluation played a critical role in this context, especially considering the unique challenges posed by its benchmark. The VOT2018 benchmark is renowned for its rigorous evaluation protocol, which includes a diverse array of challenging video sequences featuring various types of object tracking scenarios such as occlusions, rapid movements, and scale changes. Its re-initialization process, where trackers are required to recover from tracking failures, underscores the importance of not only accuracy but also the robustness and real-time performance of the tracking algorithm.

To achieve an acceptable accuracy drop, we also employed the fast fine-tuning method implemented in the Vitis-AI PyTorch quantizer, which is based on the AdaQuant algorithm [38]. This algorithm performs layer-by-layer optimization to minimize the error between the quantized and full-precision layer outputs. It involves calibrating the activations and fine-tuning the weights with a small set of unlabelled data, thereby avoiding extensive re-training. For this fine-tuning, we used five videos from the VOT2018 dataset. Meanwhile, the performance evaluation was performed on the whole dataset.

The results, as shown in Table 2, indicate that despite a slight decrease in accuracy and expected average overlap (EAO), the quantized tracker maintained an acceptable level of performance. This balance between efficiency and performance was largely attributed to the fast fine-tuning process with the AdaQuant algorithm.

Following the quantization and fast fine-tuning, we compiled the trainable components of SiamRPN++ to be run on DPU. The compilation tool used was “vai_c_xir,” part of Xilinx’s Vitis-AI suite, which compiles quantized deep learning models into a binary format suitable for execution on Xilinx hardware devices like DPU. Post-compilation, the sizes of the SiamRPN++ backbone, neck, and head components were reduced to 46 MB, 12 MB, and 1 MB, respectively, compared to 207 MB for the full-precision PyTorch model, demonstrating the effectiveness of our quantization and compilation approach.

### 3.3. Hardware–Software Implementation

Our real-time tracking system leverages the KV260 Vision AI Starter Kit from Xilinx, designed specifically for high-performance, low-latency vision AI applications and edge computing. The KV260 combines a quad-core ARM Cortex-A53 application processor with a dual-core ARM Cortex-R5 real-time processor and an FPGA fabric. This setup provides a multitude of interface options including Ethernet, USB, and HDMI, enabling a wide range of connectivity solutions.

The neural network inference of the trainable components in the SiamRPN++ tracker is executed on the Xilinx DPU B4096, a high-performance accelerator. The management of this process is facilitated by Vitis-AI Runtime (VART) [39], which offers a comprehensive low-level API for running AI models on Xilinx hardware. The programmable logic (PL) overlays and the drivers for our proposed accelerators are handled using the PYNQ 3.0 framework [40]. Operating on an Ubuntu system optimized for Kria System-on-Modules (SOMs), our setup provides a user-friendly platform with extensive software support, combining the power of DPU B4096, VART, PYNQ, and Ubuntu for efficient real-time object tracking. The overall architecture of our hardware system is depicted in Figure 2.

### 3.4. Real-Time Predictability Analysis

To assess the hard real-time capabilities of our tracker, we conducted a baseline experiment using the KV260 Vision AI Starter Kit. This experiment aimed to evaluate the predictability of each component of the tracker in a real-time scenario.

In this experiment, the trainable parts of the backbone, neck, and head of the SiamRPN++ tracker were executed on the DPU, while the preprocessing, depth-wise cross-correlation, and postprocessing were managed by a single-threaded CPU.

It is common practice in previous works (e.g., [32]) that the cross-correlation operations run on CPU. Therefore, we developed a Python script to run the depth-wise cross-correlation on CPU, using the scipy [41] library (which provides the function for single-channel cross-correlation). With the help of this function, the script applies the kernel separately to each channel of the input, yielding an output with the same number of channels as the input.

The results, presented in Table 3, highlight considerable variances in performance metrics across different components of the tracker. The ‘head’ component, in particular, shows a higher mean processing time, suggesting it as a potential bottleneck. This is further substantiated by the standard deviation and worst-case metrics, which significantly influence the overall real-time performance of the tracker.

Conversely, the ‘backbone’ and ‘neck’ components, operated on the DPU, exhibit consistent performance with minimal variation, as reflected in their low standard deviation values. This consistency is vital in real-time applications, where predictability in processing time is as crucial as speed.

In conclusion, while the DPU components (‘backbone’ and ‘neck’) demonstrate high efficiency and predictability, the ‘preprocess’ and ‘head’ components, running on the CPU, emerge as significant bottlenecks in the real-time performance of the SiamRPN++ tracker on the KV260 platform. Addressing these bottlenecks is imperative to enhance the tracker’s overall real-time predictability and performance.

## 4. Proposed Method

In this section, we present the core innovations of our work: the development of dedicated hardware accelerators to improve the real-time performance of object tracking systems. We have designed two accelerators: a padding accelerator and a depth-wise cross-correlation accelerator. These accelerators aim to address specific bottlenecks in the SiamRPN++ tracker, enhancing its efficiency and predictability in real-time environments.

We exploited high-level synthesis (HLS) in the design of the accelerators. It allows us to implement the accelerators with greater efficiency and flexibility compared with designing with hardware description language. With HLS, it is easy to adapt to the specific needs of the tracking algorithm and the embedded hardware architecture. In this work, we used Vitis HLS 2022.2 [42] as the HLS tool for optimal performance with other hardware and software from Xilinx.

### 4.1. Padding Accelerator

The padding accelerator is a critical component designed to optimize the preprocessing stage of the object tracking process. This accelerator efficiently handles padding operations, which are essential for maintaining consistent input sizes and aspect ratios for subsequent stages.

#### 4.1.1. Functional Description

Padding operations in image processing involve adding borders of specified pixel values around an image. In our case, padding is crucial for maintaining consistent dimensions for input images to the tracking model. The padding operation can be mathematically formulated as follows:$$I_{\mathrm{padded}}\left(i,j\right)=\left\{\begin{array}{cc} C & \mathrm{if}\;1\leq i\leq P_{t}\;\mathrm{or}\;\left(M+P_{t}\right)<i\leq \left(M+P_{t}+P_{b}\right)\\ \; & \mathrm{or}\;1\leq j\leq P_{l}\;\mathrm{or}\;\left(N+P_{l}\right)<j\leq \left(N+P_{l}+P_{r}\right),\\ I(i-P_{t},j-P_{l}) & \mathrm{if}\;P_{t}<i\leq \left(M+P_{t}\right)\;\mathrm{and}\;P_{l}<j\leq \left(N+P_{l}\right). \end{array}\right.$$
where


$I_{\mathrm{padded}}\left(i,j\right)$ represents the pixel value at location (i,j) in the padded image.*C* is the constant padding value, typically an average channel value.$P_{t},P_{b},P_{l},$ and $P_{r}$ are the amounts of padding added to the top, bottom, left, and right sides of the image, respectively.M×N are the dimensions of the original image.$I(i-P_{t},j-P_{l})$ indicates the pixel value in the original image corresponding to the location in the padded image, adjusted for the added padding.


In essence, this formula describes how padding is applied around an image. It adds a constant value *C* around the borders of the image, extending its size while preserving the original image data in its central part.

#### 4.1.2. HLS Design

The pseudocode for the padding accelerator (Algorithm 1) outlines the HLS logic. The process starts with the ApplyPaddingAndStream procedure, which pads the image using average channel values and streams the pixel data into the processing pipeline. This step ensures that each image frame is preprocessed uniformly, regardless of its original dimensions.
**Algorithm 1** Pseudocode for Padding Accelerator1:**Procedure** ApplyPaddingAndStream2:Apply padding with the average channel value and stream the pixel data.3:**Procedure** PaddingSingleChannel4:Apply padding to the top of the image.5:For each row in the image, apply left padding, stream actual image row data, and then apply right padding.6:Apply padding to the bottom of the image.7:**Procedure** PaddingIP8:Initialize input and output streams for each channel.9:For each channel, call PaddingSingleChannel.10:Stream out the final padded data for each channel.

In the PaddingSingleChannel procedure, padding is applied to individual image channels. This step is crucial to maintain the consistency of multi-channel images. The padding is applied to the top, bottom, left, and right sides of each channel.

Finally, the PaddingIP procedure orchestrates the overall padding process. It initializes input and output streams for each image channel and calls the PaddingSingleChannel procedure for each one. This modular approach allows for efficient parallel processing of each channel, significantly speeding up the overall padding operation.

Through this HLS design, the padding accelerator provides a fast and predictable preprocessing step, crucial for the real-time performance of the object tracking system.

### 4.2. Depth-Wise Cross-Correlation Accelerator

In our pursuit to optimize the real-time performance of object tracking on embedded systems, we identified the depth-wise cross-correlation operation as a critical component. This operation, central to the functionality of Siamese trackers like SiamRPN++, plays a pivotal role in determining the similarity between the tracked object and candidate regions. However, its intensive computational demands pose a significant challenge for real-time processing, especially on resource-constrained embedded devices.

To address this challenge, we developed a specialized hardware accelerator for depth-wise cross-correlation. This accelerator is designed to efficiently compute cross-correlation operations directly on FPGA hardware, thus offloading the computationally intensive tasks from CPU and improving the overall real-time performance of the tracking system.

We delve into the mathematical foundations of cross-correlation and depth-wise cross-correlation, followed by a detailed discussion on the HLS design and implementation of our accelerator. This approach not only enhances the performance but also contributes significantly to achieving real-time predictability in object tracking applications on embedded systems.

#### 4.2.1. Depth-Wise Cross-Correlation

As mentioned earlier, cross-correlation measures the similarity of two signals as a function of the displacement of one relative to the other. In the context of object tracking, the signals are typically feature maps extracted from images. Mathematically, the cross-correlation *R* of two discrete signals *f* and *g* is defined as follows:(1)$$R\left(f,g\right)\left[n\right]=\sum_{m=-\infty }^{+\infty }f^{*}\left[m\right]g\left[m+n\right]$$
where $f^{*}\left[m\right]$ is the complex conjugate of f[m], g[m+n] is the m+nth sample of *g*, and the sum is over all *m*.

Depth-wise cross-correlation is a variant of this concept that operates on each channel of the signals independently. If we denote $f_{c}$ and $g_{c}$ as the *c*th channels of *f* and *g*, respectively, then the depth-wise cross-correlation $R_{d}$ used in SiamRPN++ [3], in which the authors did not provide a mathematical definition, can be defined as follows:(2)$$R_{d}\left(f,g\right)\left[n,c\right]=\sum_{m=-\infty }^{+\infty }f_{c}^{*}\left[m\right]g_{c}\left[m+n\right].$$

In this case, the cross-correlation is computed separately for each channel and the results are concatenated to form the final output.

#### 4.2.2. HLS Design

To improve the predictability of the system, we propose a lightweight accelerator for the depth-wise cross-correlation operations. The pseudocode of our HLS implementation for the depth-wise cross-correlation accelerator is shown in Algorithm 2.

Our depth-wise cross-correlation algorithm begins with reading filter coefficients (template features) and pixel data (search frame features) from the memory into the streams, as ensured by the ReadFromMem function. Simultaneously, it checks the validity of the stride, height, and width of the image. Upon loading the pixels and coefficients into the streams, the Window2D function forms windows of pixels, modifying structures to accommodate new pixels. This function continues to write the window structure into a new stream after a sufficient number of pixels have been read. The next phase involves the Xcorr function, which is a single-channel cross-correlation operation. It applies the filter to each pixel window to compute an output pixel, after loading filter coefficients into a 2D array. These output pixels are then written into an output stream. The orchestrator of this data flow is the DWXcorr function. It initializes the input and output streams, and for each channel in the image, it calls the Window2D and Xcorr functions, facilitating the formation of pixel windows and subsequent filtering. Finally, it writes the output pixels to the output stream, thus completing the depth-wise cross-correlation operation. Our HLS design efficiently performs this process, leveraging the parallel processing capabilities of FPGA hardware as much as possible.
**Algorithm 2** Pseudocode for depth-wise cross-correlation accelerator.1:**Procedure** ReadFromMem2:Ensure stride, height, and width of image are valid3:Read filter coefficients from memory into coeff_stream4:Read pixel data from memory into pixel_stream5:**End Procedure**6:**Procedure** WriteToMem7:Ensure stride, height, and width of image are valid8:Read pixel data from output_stream and write it back to memory9:**End Procedure**10:**Procedure** Window2D11:Initialize LineBuffer and Window structures12:Read pixels from pixel_stream13:Shift Window and LineBuffer structures to accommodate new pixel14:After enough pixels have been read, write Window structure to window_stream15:**End Procedure**16:**Procedure** Xcorr17:Load filter coefficients into a 2D array18:**for** each pixel window in window_stream **do**19:    Apply filter to pixel window to compute output pixel20:    Write output pixel to output_stream21:**end for**22:**End Procedure**23:**Procedure** DWXCorr24:Initialize input and output streams25:**for** each channel in the image **do**26:    Call ReadFromMem to read filter coefficients and pixel data from input stream into     coeff_stream and pixel_stream27:    Call Window2D to form pixel windows28:    Call Xcorr to filter pixel windows and generate output pixels29:    Write output pixels to output stream30:**end for**31:**End Procedure**

## 5. Experiments and Results

The empirical validation of our tracking system architecture is critical to demonstrating its efficacy in real-world scenarios. This section presents our conducted extensive experiments and the results to evaluate the performance and predictability of the proposed system. We meticulously analyse the system’s behaviour, providing valuable insights into its real-time capabilities. The results underscore the benefits of our system design choices and pave the way for further enhancements in the field of embedded real-time object tracking.

### 5.1. Heterogeneous System Architecture

Our experimental setup leverages a heterogeneous system architecture, meticulously crafted to strike a balance between computational power, energy efficiency, and real-time performance. As illustrated in Figure 3, the core of the system is an ARM processor interfaced with DDR3 memory. Central to enhancing the data processing pipeline are the custom-designed accelerators: the padding and depth-wise cross-correlation (xcorr) accelerators. These specialized hardware components operate in concert with the Xilinx DPU B4096, a neural network inference accelerator, to expedite critical aspects of the tracking algorithm.

To ensure a seamless flow of data, Direct Memory Access (DMA) channels are deployed. DMAs play a pivotal role by autonomously managing data transfers between the memory and the accelerators, effectively offloading these tasks from the processor. This is crucial for reducing system latency and avoiding processor bottlenecks, thus enabling the accelerators to function at their optimal capacity. By reducing data transfer overhead, DMAs contribute to a more predictable and efficient real-time performance, which is indispensable for time-sensitive applications such as object tracking.

### 5.2. Resource Usage

The resource usage of different components of the system is crucial for understanding the efficiency of our design. The resource usage report, as detailed in Table 4, reflects the effectiveness of our system in utilizing the KV260 board’s capabilities.

From Table 4, we can deduce that the platform’s infrastructure, including the AXI interconnect, clock wizards, and DMAs, consumes a modest proportion of the total resources, suggesting a lean design. The DPU B4096 continues to utilize a considerable percentage of the resources, reflecting its role as the primary processing unit for deep learning tasks. The custom accelerators, namely the depth-wise cross-correlation accelerator (Xcorr Accel) and the padding accelerator (Padding Accel), demonstrate their efficient design by using a minimal amount of resources, which aligns with our design goals of creating lightweight yet powerful accelerators for real-time applications.

Overall, the total resource usage is around 60% for most categories, signalling a well-distributed load and the potential for scaling or integrating additional functionalities in the future. This efficient allocation of resources is indicative of our system’s capability to handle demanding tracking tasks without reaching the limits of the hardware’s capabilities.

### 5.3. Real-Time Performance Experiment Design

Our experiments were meticulously designed to evaluate the real-time performance enhancements achieved through our hardware accelerators. The baseline experiment represents a typical setup, where a single-threaded CPU is used for preprocessing, depth-wise cross-correlation, and postprocessing, complemented by the DPU handling the backbone, neck, and head of the tracker. To exploit the potential of the quad-core processor, the “multi-threading” experiment engages multi-threaded CPU cores to perform both padding and depth-wise cross-correlation operations. The “accelerators” experiment represents our advanced setup, using dedicated accelerators for both padding and depth-wise cross-correlation to alleviate the computational load from the CPU.

### 5.4. Results and Analysis

The results of running SiamRPN++ with three different setups on KV260 are shown in Table 5. In this table, the ‘total tracker’ component encompasses the entire tracking process, which includes the ‘preprocess’, ‘backbone’, ‘neck’, ‘head’, and ‘postprocess’ components. The ‘preprocess’ and ‘postprocess’ components handle the initial preparation of the input data and the final processing of the output data, respectively. The mean and standard deviation metrics for the ‘total tracker’ component in the table thus provide an overview of the real-time performance and real-time predictability of the entire tracking process.

In the ‘head’ component, where we identified the primary bottleneck, the mean processing time in the baseline is 924.76 ms. This drops significantly to 445.88 ms with multi-threading and further plummets to an impressive 54.23 ms with the use of the depth-wise cross-correlation accelerator, demonstrating the profound impact of our hardware optimizations. The drastic reduction in standard deviation from 11.08 ms in the baseline to 1.57 ms with the depth-wise cross-correlation accelerator further underscores the enhanced predictability and stability of the tracking process.

Looking at the ‘preprocess’ component, the mean processing time shows a notable decrease from 22.20 ms in the baseline to 6.06 ms with multi-threading, highlighting the benefits of utilizing multi-threading in optimizing preprocessing operations. However, the introduction of the padding accelerator further reduces the mean processing time to 2.90 ms, showcasing the profound efficiency of dedicated hardware acceleration. Most crucially, the standard deviation in this component sees a remarkable reduction from 14.26 ms in the baseline to just 0.13 ms with the padding accelerator.

The overall tracker performance sees substantial improvements as well. From a baseline mean of 1046.29 ms, we observe a reduction to 552.89 ms with multi-threading and a further decrease to 157.12 ms with accelerators, i.e., a mean total speed up with a factor 6.6. The standard deviation also reduces remarkably, from 18.87 ms in the baseline to just 1.63 ms with accelerators, indicating a more than 11 times more predictable tracking process.

The superior hard real-time behaviour of our proposed system can also be observed in the worst latency numbers: This experiment shows that our system guarantees a runtime below 187.51 ms per frame, while the baseline needs 1303.53 ms at least. In a hard real-time sense, the system can hence run almost seven times faster.

These experimental results demonstrate the profound impact of our dedicated hardware accelerators on real-time tracking performance. The substantial improvements observed with the introduction of accelerators, particularly in the ‘preprocess’ and ‘head’ components, underscore their effectiveness in significantly reducing the processing times and enhancing the overall system predictability. The dramatic decrease in both the mean processing times and the standard deviation across various components highlights the crucial role these accelerators play in achieving a more efficient and consistent real-time object tracking implementation. These findings reinforce the importance of employing specialized hardware solutions in demanding computational tasks, showcasing how dedicated accelerators can transform the landscape of real-time performance in object tracking applications.

### 5.5. Power Consumption Analysis

The analysis of power consumption forms a crucial component of our study, offering insights into the energy efficiency of our system across different configurations. The results of our power consumption experiments are shown in Table 6.

At the baseline level, utilizing a single-threaded CPU in tandem with the DPU results in a power consumption of 0.99 watts and an energy consumption of 1036 mJ to process one frame, establishing a standard for energy efficiency against which other configurations are measured.

Advancing to the multi-threading experiment, the power consumption ascends to 1.36 watts and the energy consumption drops to 752 mJ per frame. This increment in power consumption is primarily due to the increased computational load managed by the CPU when processing padding and depth-wise cross-correlation operations in a multi-threaded fashion.

Finally, the experiment employing dedicated hardware accelerators for both padding and depth-wise cross-correlation demonstrates a power consumption of 1.11 watts. Notably, this configuration achieves a more efficient power utilization compared to the multi-threading setup. Moreover, the energy consumption significantly drops to 174 mJ per frame, resulting in 5.95 times of energy saving compared to the single-threaded CPU baseline.

These findings illustrate the energy efficiency of employing dedicated hardware accelerators. The relatively marginal increase in power consumption compared to the baseline, coupled with the substantial improvements in performance and predictability, emphasizes the potential of this approach in power-sensitive embedded systems. Our system, thus, not only advances the state-of-the-art in real-time object tracking but also aligns with the imperative need for energy conservation in embedded applications.

## 6. Conclusions

In this study, we analysed and enhanced the hard real-time behaviour of a deep-learning-based object tracker, primarily focusing on leveraging hardware acceleration to address bottlenecks in real-time predictability performance. Our approach involved the utilization of Vitis-AI for quantizing the SiamRPN++ tracker and the creation of dedicated hardware accelerators for depth-wise cross-correlation and padding operations, using Vitis HLS. These components were integrated and executed on a KV260 board, forming a comprehensive system tailored for efficient object tracking.

The results from our experiments highlight the substantial improvements achieved through this approach. The introduction of the padding accelerator, in conjunction with the depth-wise cross-correlation accelerator, significantly enhanced the real-time performance of the tracking system. The accelerators not only improve processing times with more than a factor of 6 across various components of the tracker but also demonstrate a remarkable efficiency in hardware resource utilization, particularly suited for the specific demands of these operations.

A critical aspect of this study was the emphasis on improving real-time predictability, an area often overlooked in similar studies. Our results showed substantial reductions in standard deviations and worst-case execution times for processing times (rendering the execution 11 times more predictable in terms of computational latency variation), illustrating a substantial increase in the consistency and predictability of the system’s performance. This improvement is particularly vital in applications where consistent and reliable system responses are imperative.

Moreover, the power consumption analysis revealed that the use of dedicated hardware accelerators also contributes to energy conservation. The system maintained a relatively low energy consumption while delivering enhanced performance and predictability, an essential attribute for embedded and mobile applications where power efficiency is crucial.

In conclusion, this work underscores the potential of integrating hardware acceleration into real-time object tracking systems. By focusing on both speed and predictability, we have presented a promising approach that not only boosts performance but also ensures reliability and efficiency. Our methodology sets a foundation for future research and development in this field, potentially leading to more advanced and robust object tracking solutions in embedded systems.

However, it is important to acknowledge certain limitations associated with our approach. Firstly, the adoption of a heterogeneous hardware system, while beneficial for performance and predictability, could potentially increase the overall cost of an embedded system. This aspect is particularly significant as cost considerations are often paramount in the design and deployment of commercial embedded systems. Secondly, the design and implementation of the hardware accelerators detailed in our study necessitate a degree of expertise in hardware design, particularly in HLS. This requirement could pose a barrier to entry for practitioners or researchers without a background in hardware engineering.

Looking ahead, we identify several promising directions for future work to further enhance the efficacy and applicability of our approach. Firstly, the real-time predictability of the system can potentially be further improved by optimizing memory management, which is anticipated to become the new bottleneck as other components become more efficient. Additionally, a comparative analysis of our method with trackers accelerated on other forms of embedded hardware, such as embedded GPUs, is crucial for validating the efficiency and applicability of our approach across different hardware platforms. Although the accelerators designed in this study are common in many Siamese trackers, applying our method to other Siamese trackers could reveal new real-time predictability bottlenecks. Nevertheless, the methodology introduced in this paper provides a robust framework that can assist system designers in easily identifying and addressing these potential challenges, thereby facilitating the development of more reliable and efficient object tracking systems in a variety of real-world applications.

## Figures and Tables

**Figure 1 jimaging-10-00070-f001:**
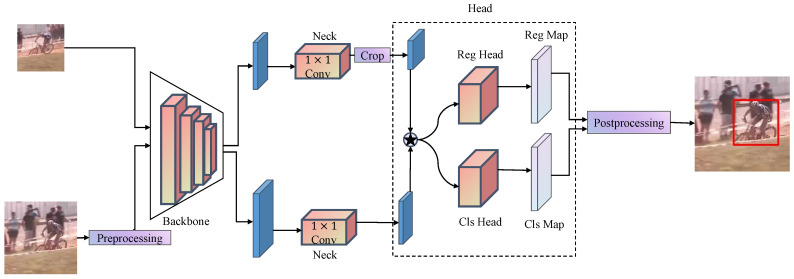
Structure of SiamRPN++ tracker. 
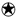
 denotes depth-wise cross-correlation. The red box denotes the predicted bounding box.

**Figure 2 jimaging-10-00070-f002:**
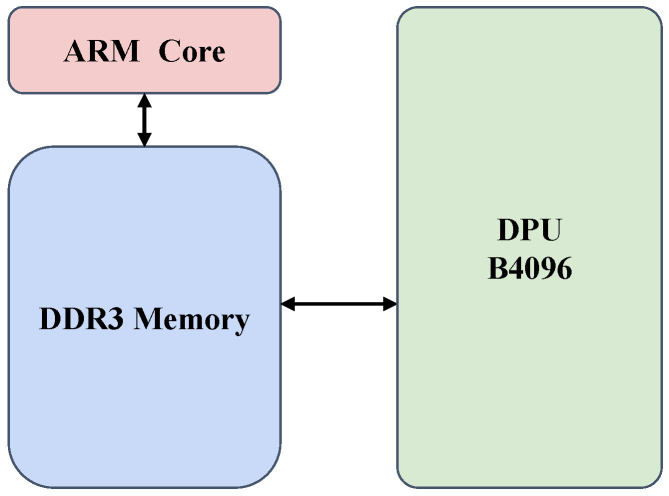
System architecture on hardware, illustrating the integration of ARM processors and FPGA fabric for efficient computing.

**Figure 3 jimaging-10-00070-f003:**
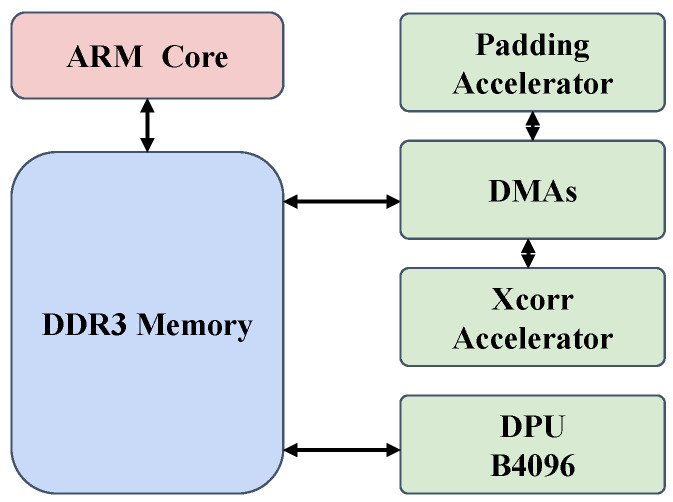
System architecture on hardware. The double-headed arrows indicate the data flow between components. The AXI interconnect and control connections are omitted for simplicity. All the components in green (Xcorr Accelerator, DMA, and DPU) are implemented on PL.

**Table 1 jimaging-10-00070-t001:** Major quantization configuration.

Quantizable Data Type	Symmetric Mode	Method	Round Method	Granularity	Scale Type
Input	Symmetric	diffs	std_round	per_tensor	poweroftwo
Weights	Symmetric	diffs	std_round	per_tensor	poweroftwo
Bias	Symmetric	diffs	std_round	per_tensor	poweroftwo
Activation	Symmetric	diffs	half_up	per_tensor	poweroftwo

**Table 2 jimaging-10-00070-t002:** Tracking performance on VOT2018.

Tracker Name	Accuracy	Robustness	EAO
Full-precision SiamRPN++	0.600	0.234	0.414
Quantized SiamRPN++	0.563	0.276	0.346

**Table 3 jimaging-10-00070-t003:** Baseline real-time performance: per frame execution latency (ms) of baseline implementation running SiamRPN++ with CPU + DPU on KV260.

	Preprocess	Backbone	Neck	Head	Postprocess	Total Tracker
Mean	22.20	90.31	5.83	924.76	3.13	1046.29
Std Dev	14.26	0.08	0.68	11.08	0.06	18.87
Best	1.75	90.12	5.44	913.69	3.04	1019.21
Worst	54.07	91.62	33.83	1156.49	4.13	1303.53

**Table 4 jimaging-10-00070-t004:** Resource usage report.

Component	LUT	LUTAsMem	REG	BRAM	URAM	DSP
Total Available	117,120	57,600	233,240	144	64	1248
Platform	16,498	3090	26,565	9	0	0
DPU B4096	50,693	6746	98,895	82	46	710
Xcorr Accel	863	78	1115	0	0	25
Padding Accel	1952	0	2088	0	0	30
Total Used	70,006	9914	128,663	91	46	765
Total Used (%)	59.77%	17.21%	55.16%	64.19%	71.88%	61.30%

**Table 5 jimaging-10-00070-t005:** Real-time Performance (per frame execution latency in ms) of running SiamRPN++ with/without our accelerators. Numbers in bold indicate the best mean and standard deviation results. Red bold numbers are used when our accelerator method achieves the best results.

Component	Metric	Baseline	Multi-Threading	Accelerators
Preprocess	Mean	22.20	6.06	**2.90**
	Std Dev	14.26	10.62	**0.13**
	Best	1.75	1.00	2.74
	Worst	54.07	30.38	3.37
Backbone	Mean	**90.31**	91.17	90.48
	Std Dev	**0.08**	0.21	0.19
	Best	90.12	90.24	90.26
	Worst	91.62	95.38	91.14
Neck	Mean	**5.83**	6.48	5.89
	Std Dev	0.68	0.56	**0.37**
	Best	5.44	5.67	5.18
	Worst	33.83	24.29	22.43
Head	Mean	924.76	445.88	**54.23**
	Std Dev	11.08	14.80	**1.57**
	Best	913.69	410.91	51.40
	Worst	1156.49	571.69	86.67
Postprocess	Mean	**3.13**	3.25	3.25
	Std Dev	**0.06**	0.19	0.09
	Best	3.04	3.38	3.11
	Worst	4.13	5.02	4.04
Total Tracker	Mean	1046.29	552.89	**157.12**
	Std Dev	18.87	18.28	**1.63**
	Best	1019.21	525.47	155.08
	Worst	1303.53	716.29	187.51

**Table 6 jimaging-10-00070-t006:** Power consumption experiment results. The red bold number indicates the best value in the energy evaluation.

Configuration	Power (W)	Mean Frame Latency (ms)	Energy (mJ/Frame)
Baseline	0.99	1046.29	1036
Multi-Threading	1.36	552.89	752
Accelerators	1.11	157.12	**174**

## Data Availability

The data presented in this study are available on request from the corresponding author due to the data is not readily available because of time limitations.

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
