# Peer review of "Enhancing Embedded Object Tracking: A Hardware Acceleration Approach for Real-Time Predictability"

_2313-433X, 2024, doi:10.3390/jimaging10030070_

Round 1

Reviewer 1 Report

Comments and Suggestions for Authors

The theme of the research is highly relevant, as hardware-accelerated trackers and their power efficiency on embedded devices are necessary in many areas. The research is well-organized, and the related papers are well-presented. The methods and results are quite clear.

However, I have a few minor suggestions:

1. The abbreviation "FPGA" is used in the abstract without an explanation (line 6).

2. Basic information about FPGAs should be provided in the Introduction section.

3. More details about the dataset and videos used in the experiments would enhance the study.

4. Tables should be placed closer to the corresponding references in the text.

5. Figures should be positioned after the references to them in the text.

6. It would be beneficial for the authors to specify if they perceive potential areas for further improvement in the methods and in which directions.

Author Response

We sincerely thank the reviewers for the time spent reading through our manuscript and for supplying valuable remarks and comments on this text. Below, you can find an overview of the changes we performed on the manuscript in order to meet the questions of the reviewers. We truly think these adaptations made the manuscript much more readable and in general increased the quality of the paper substantially. 

1. The abbreviation ”FPGA” is used in the abstract without an explanation
(line 6).
This is fixed.
2. Basic information about FPGAs should be provided in the Introduction
section.
This is fixed by adding a basic introduction of FPGA in the introduction section (text in blue).
3. More details about the dataset and videos used in the experiments would
enhance the study.
A more detailed introduction of VOT2018 is added in section 3.2.
4. Tables should be placed closer to the corresponding references in the text.
The placement of tables, figures and algorithms are adjusted according to the requirements.
5. Figures should be positioned after the references to them in the text.
Same as above.
6. It would be beneficial for the authors to specify if they perceive potential
areas for further improvement in the methods and in which directions.
A discussion of future work, including potential areas for further improvement, is added to the Conclusion section.

Reviewer 2 Report

Comments and Suggestions for Authors

Dear authors,
                    there are several editing issues to be fixed in the text. 

Line 21 Siamese networks have emerged as a leading solution, proficiently addressing….

Please introduce References describing the Siamese networks approach

Line 1  While Siamese object tracking has witnessed significant advancements, its hard real- 1time behaviour on embedded devices remains inadequately addressed.

Please justify the statement with references

Line 31-31 This is compounded by the fact that the existing literature on hardware acceleration of Siamese trackers often overlooks the intricacies involved, particularly the predictability analysis of real-time implementations

Please insert some literature citations related to the text

Line 142 Please move Figure 1 from the bottom of the page, close to the citation Figure 1

Line 158 Insert a space between FINN and the [32] reference citation

Figure 1: Please enlarge the text dimension to be easy readable

 Figure 1   Correct the text that must be inside the box

 Please move table 2 under the Line where table was cited

References should follow the format suggested by MDPI and please add the doi address where possible. Note that the references are reported in two different formats !

 More important about the topic.
The topic sound interesting, where less accuracy can speed up the tracking process thanks to the accelerators and algorithm proposed.

Section 2 is well done and well describe the state of the art

I would appreciated some more details about the real time experiments you have been running to evaluate the performances of your procedure.

Also about the results and analysis section, more details and a more deep description of the results and the experiments would be appreciated

Author Response

We sincerely thank the reviewers for the time spent reading through our manuscript and for supplying valuable remarks and comments on this text. Below, you can find an overview of the changes we performed on the manuscript in order to meet the questions of the reviewers. We truly think these adaptations made the manuscript much more readable and in general increased the quality of the paper substantially.

  1. Line 21 Siamese networks have emerged as a leading solution, proficiently addressing...
    Please introduce References describing the Siamese networks approach.
    Section 2.2 was rewritten. See the response to comment No.3.
  2. Line 1 While Siamese object tracking has witnessed significant advance-
    ments, its hard real-time behaviour on embedded devices remains inadequately addressed. Please justify the statement with references.
    Section 2.2 was rewritten. See the response to comment No.3.
  3. Line 31-31 This is compounded by the fact that the existing literature
    on hardware acceleration of Siamese trackers often overlooks the intricacies involved, particularly the predictability analysis of real-time implementations.
    Please insert some literature citations related to the text.
    Section 2.2 was rewritten to give more details of existing literature on Ob-
    ject Tracker Acceleration. This rewritten part can justify that hard real-time
    behaviour (predictability) of (Siamese) trackers on embedded devices remains inadequately addressed in the literature.
  4. Line 142 Please move Figure 1 from the bottom of the page, close to the
    citation Figure 1.
    Fixed.
  5. Line 158 Insert a space between FINN and the [32] reference citation.
    Fixed.
  6. Figure 1: Please enlarge the text dimension to be easy readable.
    Fixed.
  7. Figure 1 Correct the text that must be inside the box.
    Fixed.
  8. Please move table 2 under the Line where table was cited.
    Fixed.
  9. References should follow the format suggested by MDPI and please add the doi address where possible. Note that the references are reported in two different formats!
    Did the reviewer mean that there is underlining for some references? We
    couldn’t find the source of that because we were simply using a .bib file to
    automatically generate the references. However, the formatting issue can be of course fixed in the next version, possibly with the help of MDPI editing staff. DOI addresses are added where possible.
  10. More important about the topic. The topic sound interesting, where
    less accuracy can speed up the tracking process thanks to the accelerators and algorithm proposed. Section 2 is well done and well describe the state of the art I would appreciated some more details about the real time experiments you have been running to evaluate the performances of your procedure. Also about the results and analysis section, more details and a more deep description of the results and the experiments would be appreciated.
    We appreciate the interest of the reviewer in our exeriments. However, we
    feel all important details are in the paper, including detailed timing results and energy usage results. If the reviewer can make clear what aspects can be added to enhance the manuscript, we are happy to provide it in a final version.

Reviewer 3 Report

Comments and Suggestions for Authors

This paper introduces dedicated hardware accelerators to achieve real-time performance on embedded devices. The experimental results show the good results of this work.

However, to improve the quality of this paper in the future, the following suggestions should be taken into account.

1. In Table 2, it is recommended to perform performance tests on the complete VOT2018 dataset.

2. Whether the dedicated hardware accelerators designed in this manuscript is suitable for other Siamese network trackers.

3. It is suggested that the dataset used in the experiment or related details be supplemented in Section 5.

4. Please consider discussing and analyzing the limitations of the proposed hardware acceleration approach.

Comments on the Quality of English Language

Moderate editing of English language required

Author Response

We sincerely thank the reviewers for the time spent reading through our manuscript and for supplying valuable remarks and comments on this text. Below, you can find an overview of the changes we performed on the manuscript in order to meet the questions of the reviewers. We truly think these adaptations made the manuscript much more readable and in general increased the quality of the paper substantially.
  1. In Table 2, it is recommended to perform performance tests on the complete VOT2018 dataset.
    The original text may cause some confusion. The weights and biases were adjusted (fast-tunning) with 5 videos from VOT2018 but the performance test was done on the whole dataset. This is clarified in the new version of the manuscript by adding more information (text in blue).
  2. Whether the dedicated hardware accelerators designed in this manuscript is suitable for other Siamese network trackers.
    While only designing and benchmarking on one Siamese tracker, our methodology is universal as SiamRPN++ has typical architecture and the two bottlenecks we addressed with accelerators are commonly used in other trackers. New bottlenecks may still appear and they can be addressed with our proposed testing and designing methodology. This is explained in the newly added text in the Conclusion section (text in blue).
  3. It is suggested that the dataset used in the experiment or related details be supplemented in Section 5.
    A more detailed introduction of VOT2018 is added in section 3.2. More details can be added if necessary.
  4. Please consider discussing and analyzing the limitations of the proposed hardware acceleration approach.
    A discussion of this is added to the Conclusion section (text in blue).

Round 2

Reviewer 3 Report

Comments and Suggestions for Authors

Accept in present form

Comments on the Quality of English Language

Minor editing of English language required